

# Syndromic surveillance in companion animals utilizing electronic medical records data: development and proof of concept

Philip H. Kass[1], Hsin-Yi Weng[1,4], Mark A.L. Gaona[1], Amy Hille[2], Max H. Sydow[2], Elizabeth M. Lund[2] and Peter J. Markwell[3]

[1] Department of Population Health and Reproduction, University of California, Davis, CA, USA
[2] Banfield Applied Research and Knowledge Team, Portland, OR, USA
[3] Mars Global Food Safety Center, Huairou, Beijing, People's Republic of China
[4] Current affiliation: Department of Comparative Pathobiology, Purdue University, West Lafayette, IN, USA

Corresponding author
Philip H. Kass, phkass@ucdavis.edu

## ABSTRACT

In an effort to recognize and address communicable and point-source epidemics in dog and cat populations, this project created a near real-time syndromic surveillance system devoted to companion animal health in the United States. With over 150 million owned pets in the US, the development of such a system is timely in light of previous epidemics due to various causes that were only recognized in retrospect. The goal of this study was to develop epidemiologic and statistical methods for veterinary hospital-based surveillance, and to demonstrate its efficacy by detection of simulated foodborne outbreaks using a database of over 700 hospitals. Data transfer protocols were established via a secure file transfer protocol site, and a data repository was constructed predominantly utilizing open-source software. The daily proportion of patients with a given clinical or laboratory finding was contrasted with an equivalent average proportion from a historical comparison period, allowing construction of the proportionate diagnostic outcome ratio and its confidence interval for recognizing aberrant heath events. A five-tiered alert system was used to facilitate daily assessment of almost 2,000 statistical analyses. Two simulated outbreak scenarios were created by independent experts, blinded to study investigators, and embedded in the 2010 medical records. Both outbreaks were detected almost immediately by the alert system, accurately detecting species affected using relevant clinical and laboratory findings, and ages involved. Besides demonstrating proof-in-concept of using veterinary hospital databases to detect aberrant events in space and time, this research can be extended to conducting post-detection etiologic investigations utilizing exposure information in the medical record.

## INTRODUCTION

Surveillance provides the key linkage between naturally occurring disease or syndrome occurrence and its real-time recognition (*Henning, 2004*; *May, Chretien & Pavlin, 2009*; *Wójcik et al., 2014*). Multiple approaches to the conduct of surveillance exist, depending

in part on whether measurement of incidence is possible (as in population-based active surveillance) or not (as in hospital-based or passive surveillance). If a preponderance of evidence points to an actual disease cluster, an epidemiologic outbreak investigation should be immediately initiated: the sooner the investigation begins following a sudden increase in disease frequency, the more likely that the source of the outbreak can be identified and an intervention implemented (*Rothman, 1990*).

The last decade has seen an increase in implementation of surveillance systems both in human populations (primarily to detect pandemic infectious disease (e.g., H1N1 influenza, SARS) and bioterrorism events *Drewe et al., 2012*; *Milinovich et al., 2014*) and animal populations (*Dórea, Sanchez & Revie, 2011*). Although these systems alone do not have immediate applicability to companion animal populations, there has been interest in the United Kingdom and United States in monitoring zoonotic disease in such populations (*Day et al., 2012*; *Glickman et al., 2006*; *Halliday et al., 2007*; *Maciejewski et al., 2007*; *Shaffer et al., 2007*). While disease surveillance has been performed to a limited extent in pet animals (notably in the United Kingdom's SAVSNET and VetCompass initiatives (*Small Animal Veterinary Surveillance Network, 2015*; *Health Surveillance for UK Companion Animals, 2015*)), there have been no efforts in the last several decades to conduct real-time surveillance for syndromes or diseases in companion animals on a national scale in the United States.

Both infectious and non-infectious disease epidemics have been documented in pet animal populations in the United States over the last decade (*American Veterinary Medical Association, 2003*; *Puschner & Reimschuessel, 2011*; *Centers for Disease Control and Prevention, 2015*). One well-established example is injection-site sarcomas in cats caused by certain commonly used vaccinations. This epidemic was fortuitously recognized not through any surveillance mechanism, but through astute observation of a temporal increase in the absolute number of cases, as well as an increase in proportionate diagnostic morbidity, at a single tertiary care hospital's pathology department (*Hendrick & Goldschmidt, 1991*). Given that this epidemic was national in scope and not confined to a single vaccine manufacturer or brand, and that certain vaccines are known to increase the incidence of sarcomas two to five-fold (*Kass et al., 1993*), it is possible that it would have been detected by a surveillance system, had one been in place, that included this specific type of cancer as a diagnostic endpoint.

This underscores a singular point in surveillance methodology: that it can be difficult to distinguish "signal" (real events) from "noise" (normal or endemic background frequency of events). The strength of a surveillance system adaptive to companion animal populations thus depends on a number of factors, including: (1) the population size; (2) the magnitude of the causal effect of the risk factor; (3) the prevalence of exposure to the risk factor in the population; and (4) the baseline incidence of the outcome(s). The sensitivity of a surveillance system could be considerably improved if statistical measures were stratified by age, underscoring how active surveillance is more than real-time data mining, but also utilizes knowledge of health and disease to ask the appropriate queries and interpret the findings in a veterinary medical context.

A paradigmatic example of a point source foodborne outbreak in a pet population is the epidemic of nephrotoxicosis from ingestion of pet food adulterated with melamine

in 2007 (*Brown et al., 2007*). Although this was not initially discovered through active surveillance, had such a system focusing on syndromic and diagnostic morbidity been in place it would have had a very high probability of detecting the epidemic because: (1) the magnitude of the causal effect was large, even though the incidence of nephrotoxicosis was relatively low; (2) the prevalence of melamine in pet food diets was high; (3) the outcome was relatively specific, both as a diagnosis (acute renal disease) and as a laboratory finding (hypercreatininemia); and (4) the outcome was of sufficient severity that owners whose pets were under routine veterinary care were strongly motivated to have sought care. Again, the sensitivity of such a surveillance system could have been considerably enhanced by examining age strata, as the baseline incidence of both the laboratory finding and the diagnostic outcome would have been particular rare (i.e., prior to exposure to melamine) in younger age groups.

The goal of this research was to establish protocols devoted to near real-time surveillance of dog and cat syndrome occurrence utilizing the electronic medical records of over 700 networked primary care veterinary hospitals in the United States, which are estimated to see approximately 0.6% and 3% of the owned cat and dog populations in the United States, respectively (R Trevejo, pers. comm., 2015; *American Veterinary Medical Association, 2012*). This study's approach builds upon the classical epidemiological principle of estimating the proportional mortality ratio (PMR), which contrasts the proportion of deaths from a particular cause in an exposed group with that of an unexposed group (*Miettinen & Wang, 1981*). This construct has been extended to surveillance of adverse pharmacologic outcomes in non-hospital surveillance settings through the proportional reporting ratio (PRR) (*Rothman, Lanes & Sacks, 2004*). The current study uses the proportionate diagnostic outcome ratio (*PDOR*), a new but related metric that differs from the PMR and PRR by utilizing medical findings by health care providers instead of causes of death or adverse pharmacologic outcomes, respectively, and by treating time and geographic region as the exposures of interest. It adheres to the epidemiologic convention of favoring estimation instead of hypothesis testing, and also differs from other algorithms for signal detection. For example, the *PDOR* uses a dynamic denominator, which is different from the Recursive-Least-Square adaptive filter that uses a deterministic input signal (*Honig & Messerschmitt, 1984*). Moreover, the numerator is not included in the denominator in the computation of *PDOR*. This is different from other algorithms that implement the observed-to-expected ratio, in which the expected count is computed by including the observed count under investigation (*Kulldorff, 2015*; *Buckeridge et al., 2008*). This study reports on the development of analytic and interpretive protocols based on the *PDOR*, and their implementation to evaluate surveillance instrumental performance using two simulated outbreaks.

## MATERIALS AND METHODS

### Methodologic background for *PDOR*

The parameter of interest in relating an exposure to a health outcome is the hazard (instantaneous incidence) rate ratio parameter, defined as the ratio of the observed incidence (hazard) rate conditional on one or more covariates (X) to the potential

**Table 1 Epidemiologic measures of association and observable effect measures in longitudinal syndromic surveillance studies.**

| Measures | $T = t$ | $T = \Sigma t_i (i = 0, 1, \ldots, I; I \neq t)$ |
|---|---|---|
| Syndromic cases | $a(t)$ | $A(t_0, t_I) = \Sigma a(t_i)$ |
| Total patients seen without syndrome | $b(t)$ | $B(t_0, t_I) = \Sigma b(t_i)$ |
| Source population | $N_1(t)$ | $N_0(t_0, t_I)$ |
| Diagnostic outcome proportion | $a(t)/[a(t) + b(t)]$ | $A(t_0, t_I)/[A(t_0, t_I) + B(t_0, t_I)]$ |
| Incidence rate (hazard) | $a(t)/N_1(t)$ | $A(t_0, t_I)/N_0(t_0, t_I)$ |
| Hazard rate ratio (empirical) | $[a(t)/N_1(t)]/[A(t_0, t_I)/N_0(t_0, t_I)]$ | |
| Proportionate diagnostic outcome ratio (PDOR) | $\{a(t)/[a(t) + b(t)]\}/\{A(t_0, t_I)/[A(t_0, t_I) + B(t_0, t_I)]\}$ | |

**Notes.**

$T$, time; $i$, time points.

(expected) incidence rate in the counterfactual absence of the covariate(s). At any point in time, the incidence rate of one or more syndromes (or diseases) in a population of individuals exposed to one or more factors is contrasted with what the incidence rate would have been had the factors been absent. The incidence rate ratio (IRR) statistic is an estimate of the IRR parameter, but in a hospital-based surveillance system it is typically not possible to measure average incidence in a day (or period of days) because the population-at-risk is unknown and ill-defined. Therefore, substitute methods must be employed that allow the approximation of the incidence rate ratio statistic.

Table 1, which includes definitions of the components of the following formulas, illustrates the relationship between the IRR and the *PDOR* utilizing index ($T = t$) and referent times ($T = \Sigma t_i (i = 0, 1, \ldots, I; i < t)$), where i represents an individual time point. It is important to note that the statement: $T = \Sigma t_i (i = 0, 1, \ldots, I; i \neq t)$ can apply to any values of $t$ under partial exchangeability assumptions. However, in a (near) real-time surveillance program this is constrained to: $T = \Sigma t_i (i = 0, 1, \ldots, I; i < t)$. It can be shown (*Miettinen & Wang, 1981*) that the *PDOR*, $\{a(t)/[a(t) + b(t)]\}/\{A(t_0, t_I)/[A(t_0, t_I) + B(t_0, t_I)]\}$, can be used to estimate the hazard rate ratio, $[a(t)/N_1(t)]/[A(t_0, t_I)/N_1(t_0, t_I)]$ when: $[a(t) + b(t)]/N_1 = \{A(t_0, t_I) + B(t_0, t_I)\}/N_0(t_0, t_I)$; note that $a(t)$ and $b(t)$ represent syndromic cases and non-syndromic patients at time $= t$, $A(t_0, t_I)$ and $B(t_0, t_I)$ represent the total number of syndromic cases and non-syndromic patients seen between times $t_0$ and $t_I$, respectively, and $N_1$ and $N_0$ represent the size of the source populations at times $t$ and time period $(t_0, t_I)$, respectively. This translates into the assumption that even in the presence of an outbreak, the overall incidence of visiting a hospital for a syndromic diagnosis among the source population of dogs and cats is the same at index and referent times. This requires a "counterbalancing" of incidence: as the incidence of diagnosing a particular syndrome at $T = t$ increases, there must be a commensurate *decrease* in the incidence of diagnosing *other* syndromes at $T = t$. This assumption is reasonable only when the syndrome of interest under surveillance is rare (e.g., approximately less than 5%) compared to other diagnoses. Based on the diagnostic outcome proportions (admittedly, not incidence rates) observed, this may be a reasonable assumption in many cases (with the exception, perhaps, of older age group(s)). Achieving such rarity is also facilitated by the kind of patients that this study's hospitals typically see, because they emphasize preventive care: in the hospital's population, 27.5% of dogs and 25.2% of cats (2014 internal data) were

reported to be healthy, in contrast to 6.8% and 9.5% for dogs and cat respectively reported as healthy in the private companion animal practice population, and their patients' ages are relatively younger than those of the potential population of patients (*Trevejo, Yang & Lund, 2011*; *Lund et al., 1999*). This implies that the sample age distribution might not be representative of age distribution in the source population; thus, stratification by age is indicated.

If the rarity assumption holds, then the lesser assumption that the proportion of patients seen without a particular syndrome of interest remains relatively constant over time, also holds: $b(t)/N_1(t) = \Sigma b(t_i)/N_0(t_0, t_I)$. This should be reasonable if there are no secular trends in syndromic incidence, which appeared to be empirically true with most hospital data examined prior to commencing this study. The closer $T = t$ is to $(t_0, t_I)$, the more reasonable this assumption becomes, and the more closely the *PDOR* corresponds to the hazard rate ratio.

## Background for syndromic definitions

A workshop was convened that included external academic experts in epidemiology, nutrition, toxicology, infectious diseases, internal medicine, food safety, and clinical pathology in order to establish a set of syndromes optimal for conducting foodborne disease surveillance in companion animals; none were actually involved in the design of this research or in the preparation of this manuscript. An evaluation of electronic medical nomenclature and data fields was done to identify differences between the data desired from the workshop and the data available in the hospital network database. Thirty-seven syndromic components (i.e., clinical findings, including laboratory results) were selected for further study because these would have been the most likely to have been recognized in past foodborne disease outbreaks in pet animals, from which the following 10 were adopted for proof-of-concept in the current study: anorexia, elevated alanine aminotransferase (ALT), elevated serum calcium, elevated creatinine, diarrhea, lethargy, a *Salmonella*-positive fecal sample, seizures, urolithiasis, and vomiting.

## Information technology: data acquisition and transfer

An automated and efficient system of data transfer was required for the near real-time design of this system. The following were system analysis and design considerations judged to be of critical importance towards the success of this project: ubiquitous data mapping, high performance, high availability, storage capacity, and timely reporting. To attain "high availability," we sought to design a system that could be adapted for data input from virtually any source. The system thus developed, called "Aberrant Diagnostic Outcome Repository in Epidemiology" (ADORE), was envisioned as a potential center of a future constellation of potential separate or simultaneous information technology sources, including universities, diagnostic laboratories, institutes, government agencies, and private practices. In this research, however, only a single source was utilized.

Eight tables containing relevant information were queried in the Banfield database. Each was searched for codes specific to each syndrome. If the syndrome was found in one or more tables, then it was marked as present. Data transferred were restricted to de-identified

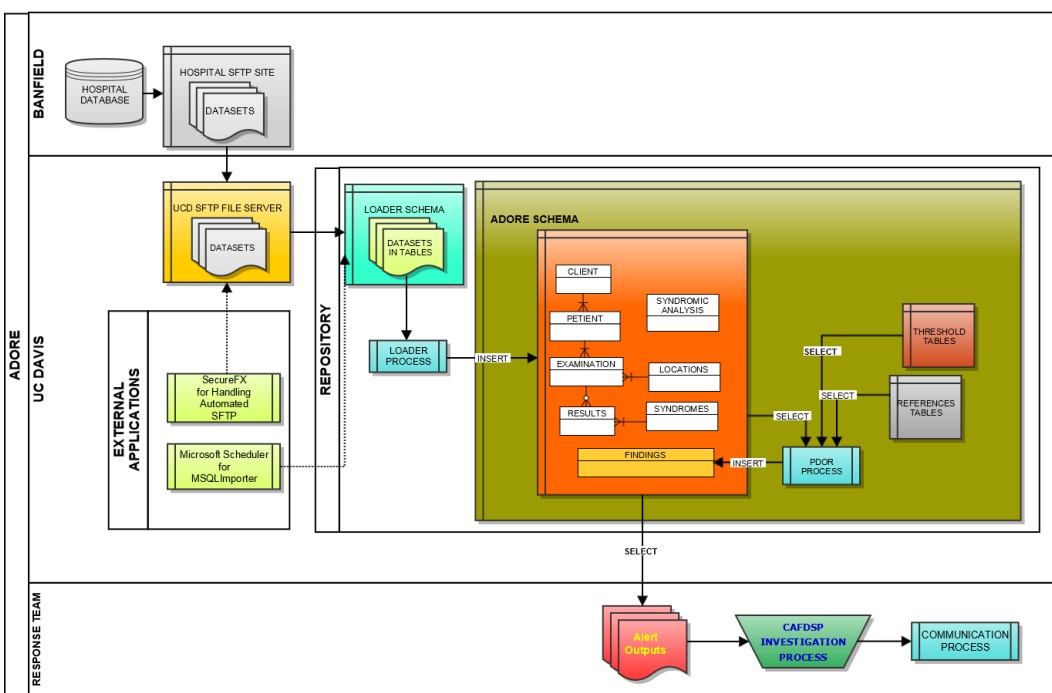

**Figure 1** Schematic of the loader process used to transfer data from a hospital database to a repository used for syndromic surveillance analyses.

numbers linking owners and patients, patient demographic information, hospital, number of encounters, and the ten syndromes analyzed in this manuscript. All hospitals used the same electronic medical record software.

We initially utilized Microsoft Windows-based programs, including Eclipse for Java development, MySQL for database architecture, and the UC Davis SmartSite curricular repository for direct data transfer between the information technology center for the hospital network and the University of California, Davis (UC Davis). However, due to security challenges, an alternative approach to real-time transfer was developed. This entailed creating custom scripts at the hospital network's information technology center based on data mapping of the ADORE system and utilizing a secure FTP site; software utilized included SecureFX and MySQL Importer. Data was via secure FTP application transferred from flat (pipe (|) delimited text) files provided by Banfield based on queries from their datasets to our UC Davis server. MySQL Importer tool was utilized to transfer the data from the flat files into tables located in the LOADER Schema. A process called Loader was run against the raw data in these tables, which transformed the data based on the validation and transformation criteria for the project, and inserted into the appropriate ADORE tables or were flagged as exceptions for addressing (Fig. 1).

Key tables were created at UC Davis for data loading, which included the following database tables: LOADER, EXAMINATIONS, PETIENTS (i.e., pet patients), CLIENTS, LOCATIONS, HOSPITALS, RESULTS, and FINDINGS tables. It was agreed upon that multiple-day sets of data (seven days) would be provided to the UC Davis team to allow

data to be transferred in a relatively short period of time. The datasets were transferred to the UC Davis Repository Monday through Friday each week starting on April 25, 2011 and continuing through June of 2011, which allowed approximately six months of data from 2010 (i.e., May 1, through October 20, 2010) to be transferred. In total, over 4.2 million patient records with data were imported and used for retrospective surveillance.

An external Scientific Advisory Board, comprised of experts in epidemiology, statistics, and public health, created two foodborne disease outbreak scenarios that were embedded into the system data without disclosure to the UC Davis team. Such blinding was deliberate, in order simulate how the UC Davis team and the ADORE system would perform if surveillance was prospectively implemented in real-time. The outbreaks were designed with the intention of determining whether the team and system could detect them, how long it would take to detect them following their onset, and to measure their magnitude upon detection (*Centers for Disease Control and Prevention, 2001*). The outbreak data, consisting of simulated medical records of patients, were created by considering the following factors:

A. Historical hospital data of the usual prevalence of clinical signs.
B. A causative agent or chemical.
C. The contaminated food product.
D. The proportion of patients consuming the contaminated food product.
E. The proportion of patients consuming the contaminated food product exhibiting the syndrome.
F. The number of hospitals in the affected region.
G. A susceptible patient population (e.g., species, age).
H. The clinical syndrome appropriate to the food contaminant.
I. The production/distribution pattern of the contaminated food, including the amount produced, the proportion of bags affected, the geographic food distribution, and the average shelf life.
J. The incubation/latency period before syndromic occurrence.

### *PDOR* procedure implementation
#### *Temporal cluster detection*
The specific adaptation of the temporal *PDOR* procedure used in the current study compared the proportion of patients seen at network hospitals on a particular day that were positive for a particular clinical or laboratory finding with the average proportion of patients positive for the same finding over a seven-day baseline period that ended three months earlier. The use of a seven-day period (which can be modified in the algorithm) allowed for within-week cyclicity of diagnoses, and the use of a three-month lag time (which can also be modified in the algorithm) was suited for a slowly developing foodborne outbreak. The premise behind the temporal cluster detection method was that the proportion of patients diagnosed with individual clinical or laboratory findings should not meaningfully change over a three month period (i.e., there are no seasonal trends), and that the daily (unobservable) incidence rate over a seven-day period (which is not equivalent to the rate of presentation to a veterinary hospital) was constant. These analyses

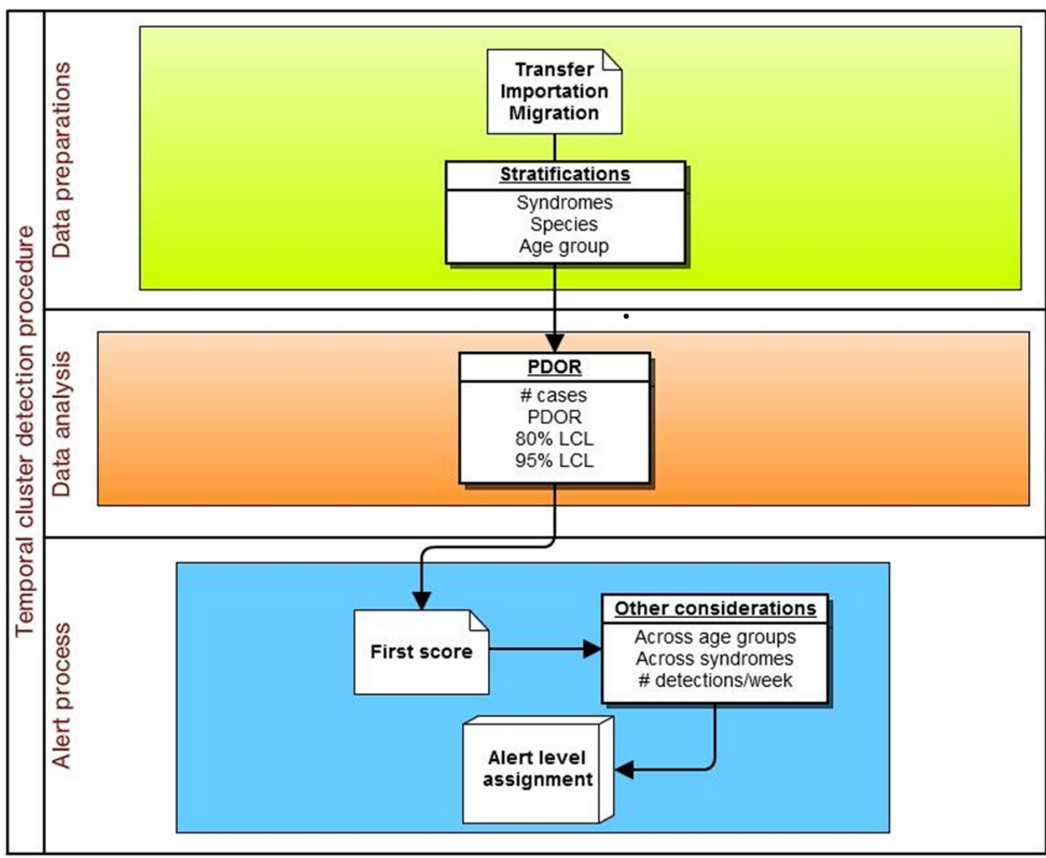

**Figure 2  Schematic of temporal analysis using the Proportionate Diagnostic Outcome Ratio procedure.**

were performed separately by species (dogs and cats) and in four age strata (<3 years, 3–7 years, 8–12 years, and 13 or more years), as well as for all ages combined (Fig. 2). The following details how the procedure was implemented.

### Variables and equations

A. Number of cases ($C_i$) on current ($i$ th) date. Note that if the same syndromic finding had been noted for the same animal multiple times at the same hospital visit, and if any of these syndromes fell outside the defined threshold range for that finding, this animal was classified as a case. Each animal was counted only once in the analysis for that hospital visit.

B. Number of hospital visits on current date ($N_i$).

C. Diagnostic Outcome Proportion of current date ($DOP_i$) $= C_i/N_i$.

D. Baseline (referent) time window (BW).

E. Lag time (l): between current date and the latest date of the baseline time window.

F. DOP of baseline ($DOP_B$), computed as the ratio of the total number of cases used for the baseline ($C_B$) to the number of patients seen used for the baseline ($N_B$) in the

specified time interval defined by l and BW:

$$\frac{\sum\limits_{t\,=\,i-(l+BW)}^{t\,=\,i-l} C_t}{\sum\limits_{t\,=\,i-(l+BW)}^{t\,=\,i-l} N_t}$$

where $C_t$ and $N_t$ are the number of cases and number of hospital visits, respectively, on date $t$, and $i$ is a time point. If no case occurred during the baseline time window (i.e., $C_B = 0$), $C_B$ was set to 1.

G. $PDOR = DOP_i/DOP_B$.

H. The following is the equation for the percent confidence limits (CL) for $PDOR$:

$$\exp\left(\ln\,(PDOR) \pm D \times \sqrt{\frac{1-DOP_i}{C_i} + \frac{1-DOP_B}{C_B}}\right)$$

where exp () is the exponential function; ln is the natural logarithmic transformation; D = 1.28 for an 80% CL and 1.96 for a 95% CL. Note that if $PDOR = 0$, the CL = 0. An 80% lower confidence limit (LCL) was selected to increase detection sensitivity in the early stages of an epidemic.

I. If a particular date was missing (e.g., due to hospital closure, such as on Christmas day) in the baseline time window, then the baseline time window was set back one more day, so that the $[i - (l + BW) - 1]$ th day was used in the computation for $DOP_B$.

J. If seven out of seven days in the baseline window all had an 80% LCL > 1, the entire week was replaced with the previous baseline window.

K. When a temporal cluster was detected, the equation for an exponentially weighted moving average (MA) was used to smooth plots of the DOP: $E_{t-1} + \alpha(O_t - E_{t-1})$, where the $E$'s are MA values and $O$'s are observed values. $\alpha$ (weighting factor) is estimated by using $2/(1+K)$, where $K$ = the number of days in the moving average (i.e., $K = 7$ for a weekly moving average). The initial value was set as $E_0 = O_0$ (or an average of a time period in the past).

### Spatial cluster detection

The spatial cluster $PDOR$ procedure compared geographic regions in the United States using two levels of granularity: US Census Divisions (USCD) ($n = 9$) and Metropolitan Statistical Areas (MSA) ($n = 39$) (Fig. 3). Each USCD's and MSA's DOP was compared with the average of the other USCDs or MSAs, respectively. The spatial cluster procedure involved two queries:

(1) Comparing among the spatial locations to identify spatial clusters (a "first query").

(2) Locations that exceeded the predetermined alert level from the first query then had a "second query" procedure performed within each of the detected spatial locations from the first query to examine whether there was a within-location temporal cluster.

The following are details of how the procedure was implemented (Fig. 4):

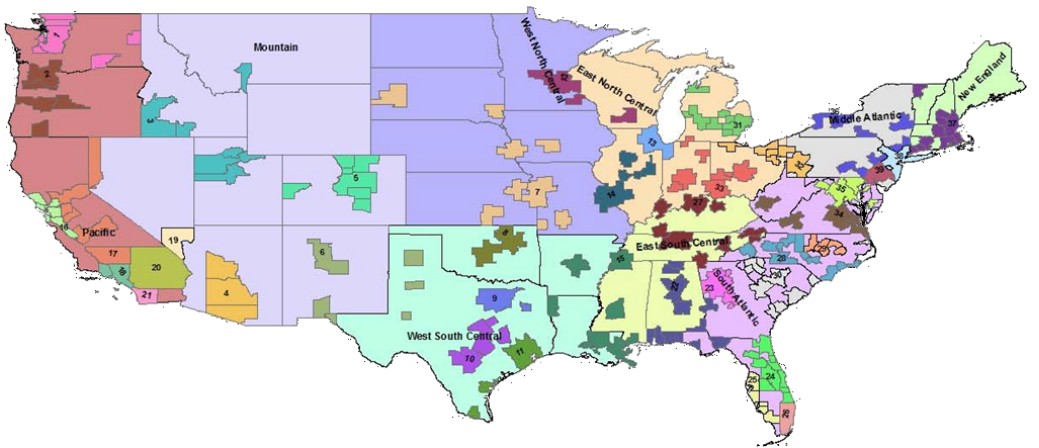

**Figure 3** **Nine US Census Divisions and 39 Metropolitan Statistical Areas (MSA) used with _PDOR_ procedure.** Census divisions covers the United States, while MSA encompass regions only where hospitals are located.

### First query variables and equations

A. USCD assignments were based on the client's home address, and MSA assignments were based on the hospital location.

B. Each clinical and laboratory finding was analyzed separately. Analysis by USCD was conditional on species and age group, as described above for temporal cluster detection. Analysis of MSA was stratified by species but not by age group.

C. $PDOR_i = DOP_i/DOP_{(A[-i])}$, where DOPs were the Diagnostic Outcome Proportions as described above. Let A = the total number of geographic units (USCD or MSA). $PDOR_i$ was the ratio of DOP at a particular USCD/MSA ($i$) and DOP of all other USCDs/MSAs ($A[-i]$). $DOP_{(A[-i])}$ was computed as total number of cases on a current date among all USCDs/MSAs except for location $i$ divided by the total number of hospital visits on a current date in the same locations. If the number of hospital visits for a particular USCD/MSA ($i$) (i.e., the denominator of DOP$i$) was zero, $DOP_i$ and $PDOR_i = 0$.

D. The equation for % confidence limits (CL) for _PDOR_ is:

$$\exp\left(\ln(PDOR_i) \pm D \times \sqrt{\frac{1-DOP_i}{C_i} + \frac{1-DOP_{A[-i]}}{C_{A[-i]}}}\right)$$

where exp () is the exponential function; ln is natural logarithmic transformation; D = 1.28 for 80% CL and 1.96 for 95% CL; $C_i$ is number of cases on current date at spatial unit $i$ and $C(A[-i])$ is total number of cases among all locations except for spatial unit $i$. If the same syndromic finding had been noted for the same animal multiple times at the same hospital visit, and if any of these syndromes fell outside the defined threshold range for that clinical finding, this animal was classified as a case. Each animal was counted only once in the analysis for that hospital visit. Furthermore, if no case occurred in baseline locations (i.e., $C_{A[-i]} = 0$), then $C_{A[-i]} = 1$. If $PDOR = 0$, then CL = 0.

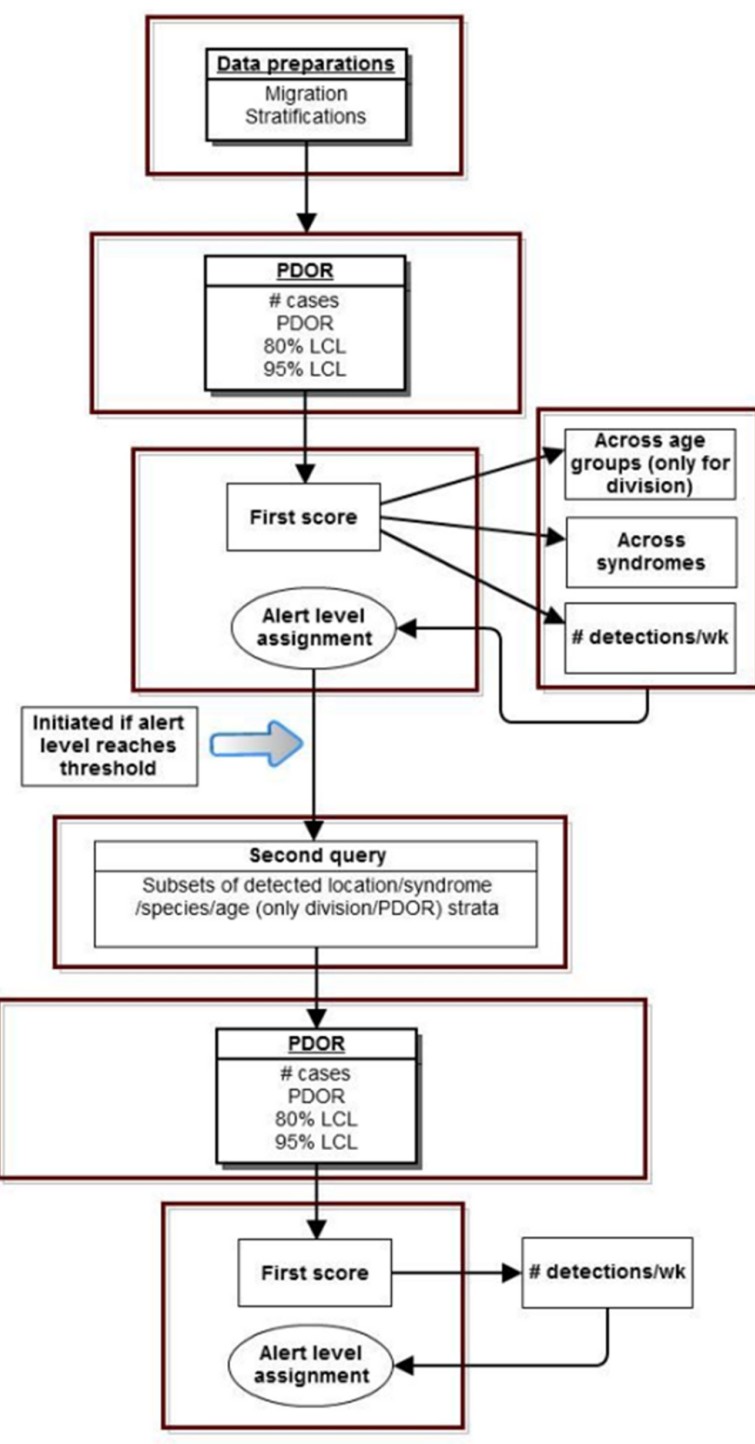

**Figure 4** Schematic of spatial analysis using the Proportionate Diagnostic Outcome Ratio procedure.

**Table 2  PDOR scoring system for temporal analyses and spatial analyses (first query).**

| Statistic | Cutoff value | Score |
|---|---|---|
| PDOR[a] | <1.25 | 0 |
|  | 1.25–1.99 | 3 |
|  | 2–2.99 | 5 |
|  | 3–3.99 | 7 |
|  | ≥4 | 9 |
| Number of cases | <3 | 0 |
|  | ≥3 | 2 |
| 80% LCL[b] | ≤1.1 | 0 |
|  | >1.1 | 2 |
| 95% LCL[b] | ≤0.8 | 0 |
|  | 0.8–1.1 | 2 |
|  | ≥1.1 | 4 |
| Across age groups | Sum[c] ≥ 5 in two or more age groups | 3 |
| Across syndromes | Sum[c] ≥ 5 in two or more syndromes | 2 |
| Across days | Sum[c] ≥ 6 on three or more days in a week | 4 |

**Notes.**
[a] Proportionate diagnostic outcome ratio.
[b] Lower confidence limit of a PDOR.
[c] Summed score of PDOR, Number of cases, 80% LCL, and 95% LCL.

### Second query variables and equations

The second query followed the steps described for the PDOR procedure for temporal cluster detection nested within each spatial cluster where there was evidence of a cluster detected on the first query. The current threshold for entering the 2nd query was set for these analyses to be $C_i \geq 5$ and $PDOR_i \geq 1.25$.

### The alert model

Each day almost 2,000 analyses were run, requiring an efficient mechanism for identifying evidence of true positives (real outbreaks) while minimizing the number of false positives, equivalent to increasing both the sensitivity and specificity of the PDOR procedure. To conduct what was essentially an efficient screening process, we developed a five-stage color-coded alert system: green (level 1) was normal, with successively higher levels: blue (level 2), yellow (level 3), orange (level 4), and red (level 5). The stronger the evidence was for temporal or spatial clustering, the higher the alert level.

Statistics used in determining alert levels included PDOR, total number of cases, and lower limits of 80% and 95% confidence intervals of a PDOR. The scoring system with cutoff values for each statistic is summarized in Table 2. The choice of cutoff values presented here was based on expert opinion and a consensus among project team members. The ADORE system, however, allows users to choose cutoff values and scores.

The scoring was first applied to each syndrome, species, and age group combination, and within each combination a sum was computed. The system then evaluated summed scores across strata. If a sum ≥ 5 occurred in two or more age groups or syndromes, or a sum ≥ 6 occurred on three or more days in a week, additional scores were added (Table 3).
**Table 3** PDOR scoring system for spatial analyses (second query) with five or more cases and PDOR ≥ 1.25 on the spatial analysis first query.

| Variables | Levels | Score | Notes |
|---|---|---|---|
| Statistics used to determine score for first step (score ≥ 5 is positive) | | | |
| *PDOR* | ≥4 | 9 | |
| | ≥3 | 7 | |
| | ≥2 | 5 | |
| | ≥1.25 | 3 | |
| | <1.25 | 0 | |
| 80% LCL | >1.1 | 2 | |
| | ≤1.1 | 0 | |
| 95% LCL | >1.1 | 4 | |
| | >0.8 | 2 | |
| | ≤0.8 | 0 | |
| Additional scores added to the first score to determine alert level | | | |
| Number of days detected within one week | ≤3 | 0 | Applied to same syndrome/age stratum/species. |
| | ≥3 | 4 | Score applies when sums are ≥5 on ≥3 days in a one week period. |

This final score was used to determine alert levels: green: ≤ 8; blue: 9–12; yellow: 13-15; orange: 16–18; and red: ≥ 19. Because no age stratification was applied in MSA analysis, the additional score for a sum ≥ 5 occurred in more than one age group was not applied. Therefore, the cutoff values for alert levels were reduced to: green: ≤ 6; blue: 7–10; yellow: 11–13; orange: 14–16; and red: ≥ 17.

## RESULTS

The first aberrant event detected that was found to be a highly plausible outbreak yielded an initial alert on May 15, 2010 (Figs. 5–7). The report generated for the week of May 9–May 13, 2011 (pertaining to the dates May 8–May 16, 2010) described an event occurring in dogs of all age groups in the Pacific USCD, and particularly in MSAs 16–18, 20, and 21 within the Pacific USCD. The clinical finding was diarrhea. Had this been real-time surveillance, we would have notified the hospital personnel on May 18, 2010, the date that we would have strongly believed that this was an actual outbreak. This situation continued to be monitored through June 13–June 17, 2011 (pertaining to dates July 4–July 18, 2010). The alerts continued unabated through July 10, 2010, after which the outbreak appeared to have resolved. The spatial proximity of the MSAs involved in this aberrant event was striking and strongly suggestive of a point source outbreak. There was no compelling evidence that this outbreak spread to other regions outside the MSAs identified above.

Following our submission of a final report of this discovery, the Scientific Advisory Board revealed to the UC Davis team that the aberrant event detected was in fact a provocative challenge (simulated foodborne outbreak). Simulated medical records of dogs in all age categories with diarrhea were randomly assigned to 81 hospitals in California and incorporated into the daily data transfer (Table 4). The outbreak consisted of the following conditions:
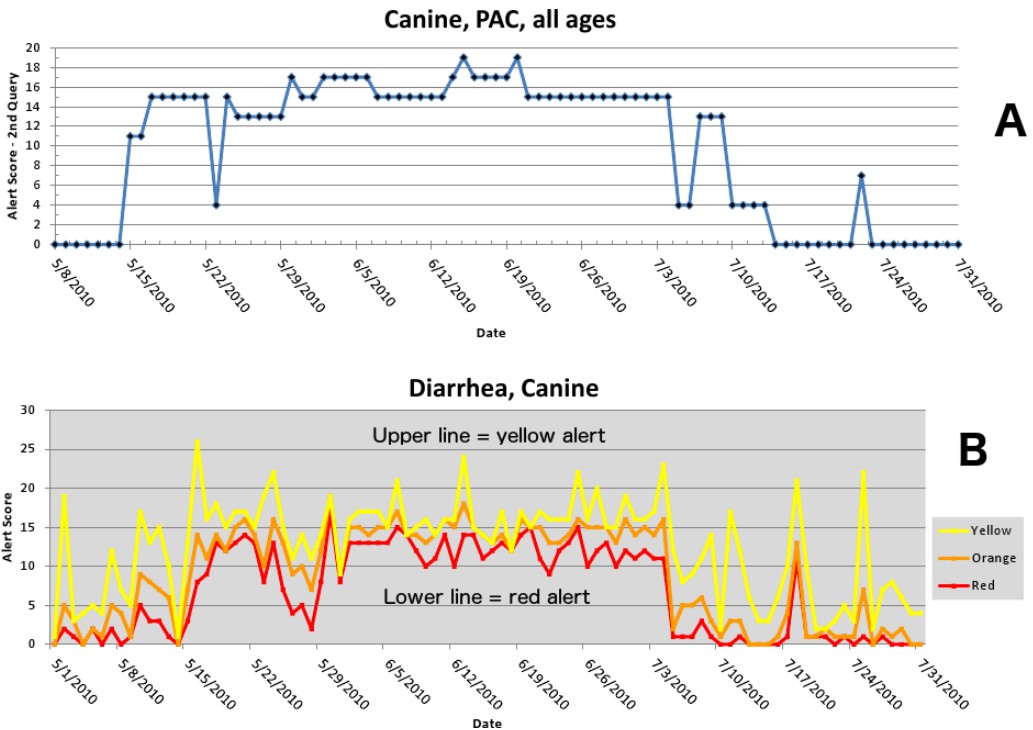

**Figure 5 Simulated canine infectious agent outbreak in California.** (A) Alert scores for diarrhea in Pacific Census Division by date (spatial analysis, second query). (B) Number of alerts above baseline, by alert color over time (temporal analysis).

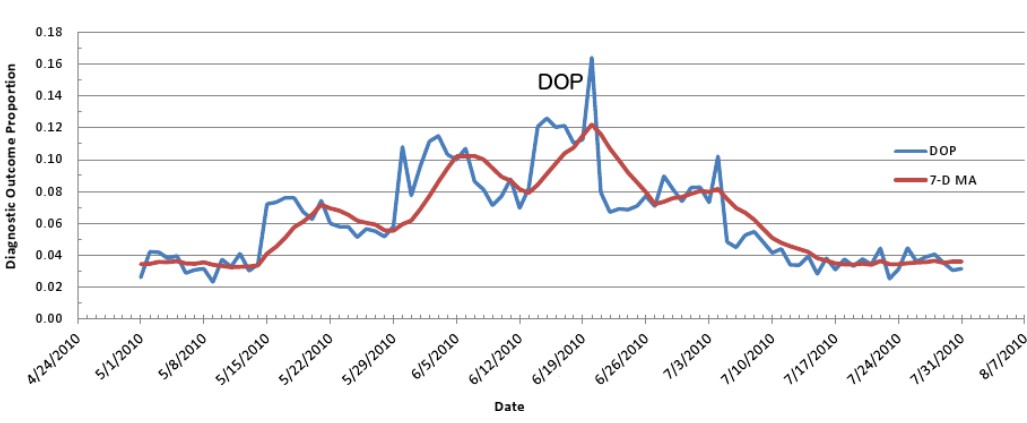

**Figure 6 Simulated canine infectious agent outbreak in Pacific Census Division.** Graph shows diagnostic outcome proportions (DOP) using diarrhea. Red line shows seven day moving average (spatial analysis, second query).

A. Cause: infectious agent causing acute gastrointestinal disease.
B. Contaminated product: dry dog food made by Company "X" in their Reno, NV plant.
C. Susceptible population: This food is marketed to all ages, breeds, and sizes of dogs.
D. Attack rate: 8% of hospital network patients consumed this food, and 11% of those who consumed it were clinically affected.
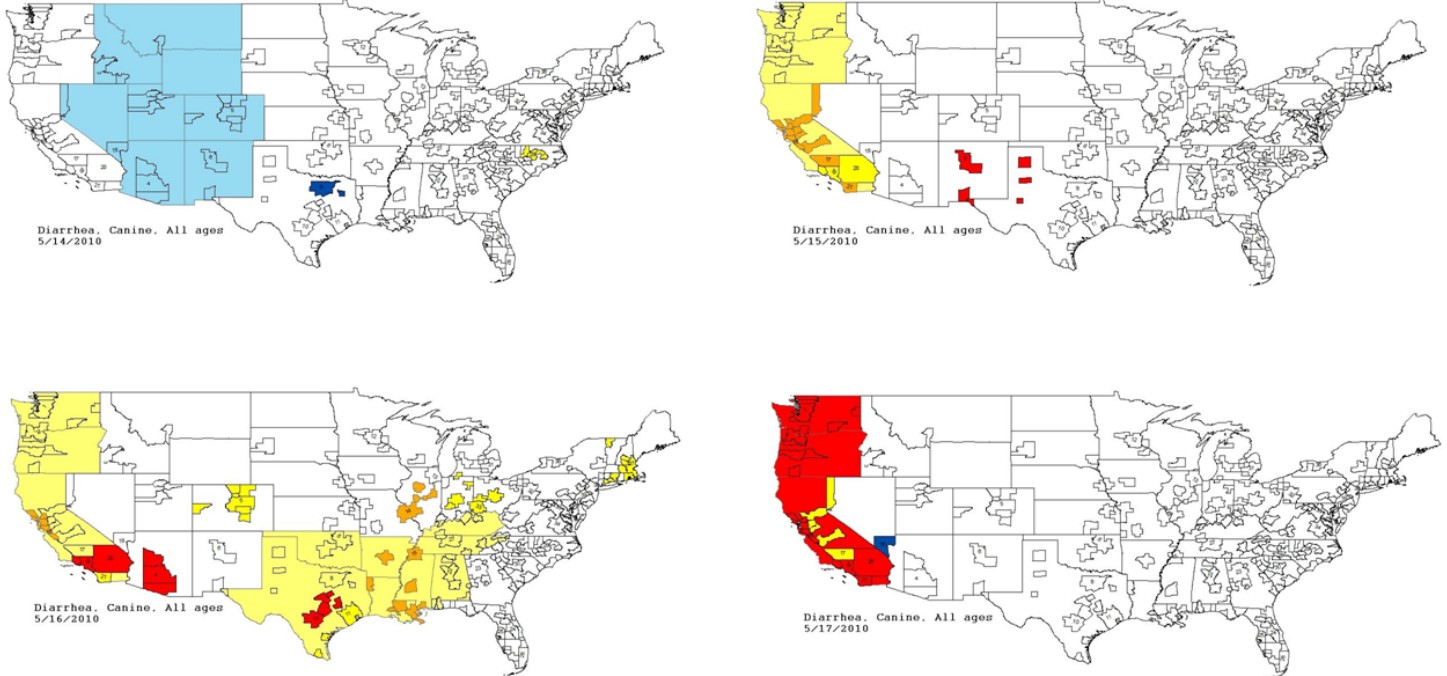

**Figure 7  Spatial analysis of gastrointestinal disease outbreak over four days.** Pacific Census Division begins with no alert (white); then yellow and orange; then yellow, orange, and red; and finally predominantly red.

**Table 4  Numbers used to simulate the first aberrant event: an outbreak of diarrhea caused by an infectious agent.** Normal average daily prevalence of diarrhea is 3.4%.

| Week | Percent of dogs with diarrhea caused by infectious agent | Total percent of dogs with diarrhea | Total number of dogs per week with diarrhea at each hospital |
|---|---|---|---|
| 1 | 5.2 | 8.6 | 12 |
| 2 | 2.8 | 6.2 | 9 |
| 3 | 10.2 | 13.6 | 19 |
| 4 | 6.5 | 9.9 | 14 |
| 5 | 13.0 | 4.7 | 7 |
| 6 | 5.7 | 9.1 | 13 |
| 7 | 6.5 | 9.9 | 14 |
| 8 | 1.7 | 5.1 | 7 |
| 9 | 0 | 3.4 | 5 |

E. Finding for clinically affected animals: diarrhea.
F. Product/distribution information: 1,462 of 2,750 tons (53%) of food produced per five days in the plant were distributed to the State of California. The hospital network operates 81 hospitals in California, and the assumption was that the diets were equally distributed throughout the state.
G. The average number of dogs seen at each hospital per day was 20, and the usual proportion of dogs seen with diarrhea was 3.4%.

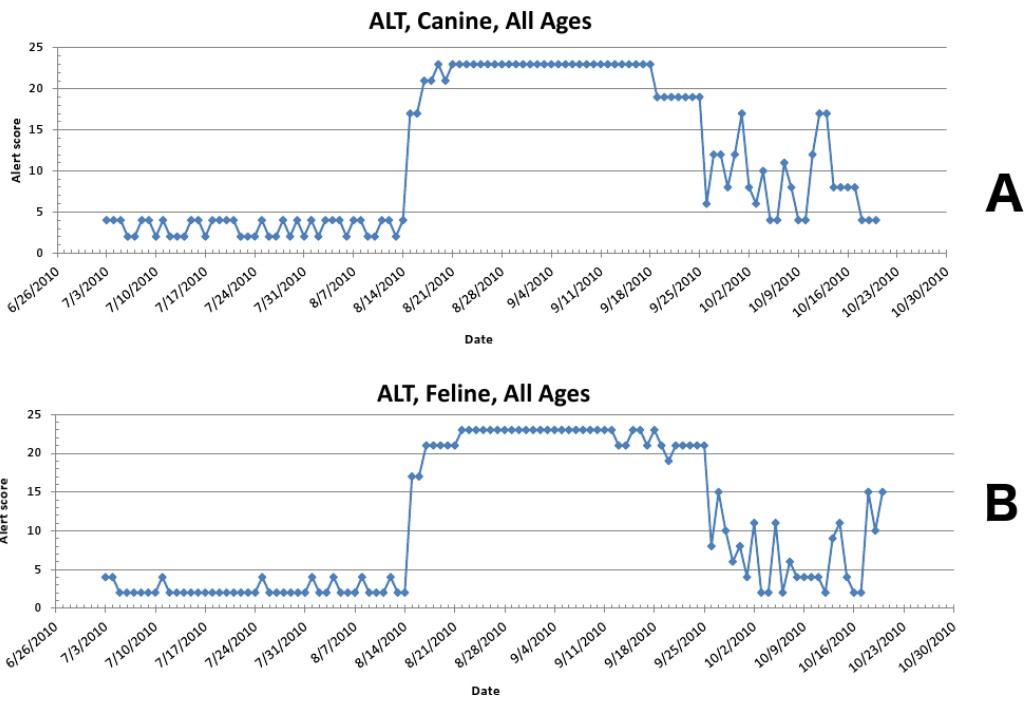

**Figure 8** **Simulated aflotoxicosis outbreak in the US using elevated alanine aminotransferase (ALT) as the clinical finding.** Graphs (canine (A), feline (B)) show alert scores by date (temporal analysis).

H. Dates: the challenge data were embedded starting on May 15, 2010 and continued through July 17, 2010.

The second aberrant event found to be a highly plausible outbreak yielded an initial alert for August 15, 2010 (Figs. 8–11). Our report for the week of June 27–July 1, 2011 (pertaining to the dates July 31–August 19, 2010) described the clinical and laboratory findings as elevated ALT, anorexia, and lethargy. The average *PDOR* for this time was 3.7 ± 2.0 for ALT, 4.4 ± 1.7 for anorexia, and 3.7 ± 1.2 for lethargy. Both dogs and cats in all age groups were affected. This event was not restricted to a single USCD, but appeared to be national in scope. That is, the temporal alerts were far more compelling in identifying this event than the spatial alerts. The constellation of clinical and laboratory findings was compatible with a hepatotoxic contaminant (such as an aflatoxin), and the enormous rise in the PDORs would have caused us to define this as an actual outbreak by August 18, 2010.

This situation continued to be monitored through the July 18–July 22, 2011 reporting period (pertaining to the dates September 24–October 20, 2010). The red alerts continued unabated through September 26, 2010, after which the outbreak appeared to have resolved.

Following our submission of the final report, the Scientific Advisory Board revealed to the UC Davis Team that the aberrant event detected was in fact a provocative challenge (simulated foodborne outbreak). Simulated medical records of cats and dogs in all age categories with clinical signs typically seen in patients with hepatic disease (e.g., elevated ALT, anorexia, and depression) were randomly assigned to hospitals throughout the US

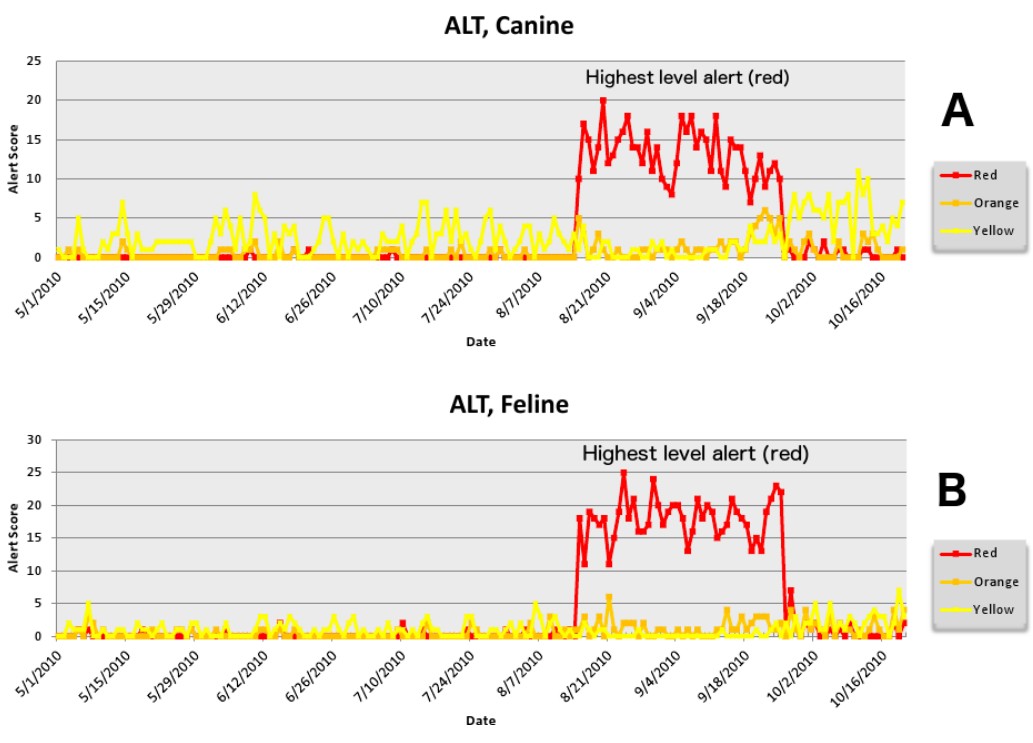

**Figure 9  Simulated aflotoxicosis outbreak using alanine aminotransferase (ALT) as syndrome.** Graphs (canine (A), feline (B)) show number of alerts above baseline, by alert color over time (temporal analysis).

and incorporated into the daily data transfer (Tables 5 and 6). The outbreak was constituted by the following conditions:

A. Causative agent: aflatoxin.
B. Contaminated product: all formulas of dry cat and dog food made in one central plant in the Midwest USA during a two-week period. Cornmeal used in the production of food was contaminated with aflatoxin.
C. Susceptible population: all ages, breeds, and sizes of dogs and cats.
D. Attack rate: 7.3% of dogs and 6.7% of cats were fed this diet, and 30% of those who consumed it were affected.
E. Clinical and laboratory findings for affected animals: anorexia, depression (lethargy), and icterus (ALT > 100 in cats, ALT > 118 in dogs).
F. Product/distribution information: the food was distributed from the single plant to the entire country.
G. The average number of cats and dogs seen at each hospital per day was 5 and 20, respectively. The usual proportion of cats and dogs seen with elevated ALT was 2.5% and 2.7%, respectively; with anorexia 4.3% and 2.5%, respectively; and with depression 1.9% and 3.5%, respectively.
H. Dates: the challenge data were embedded starting on August 15 running through September 26, 2010.

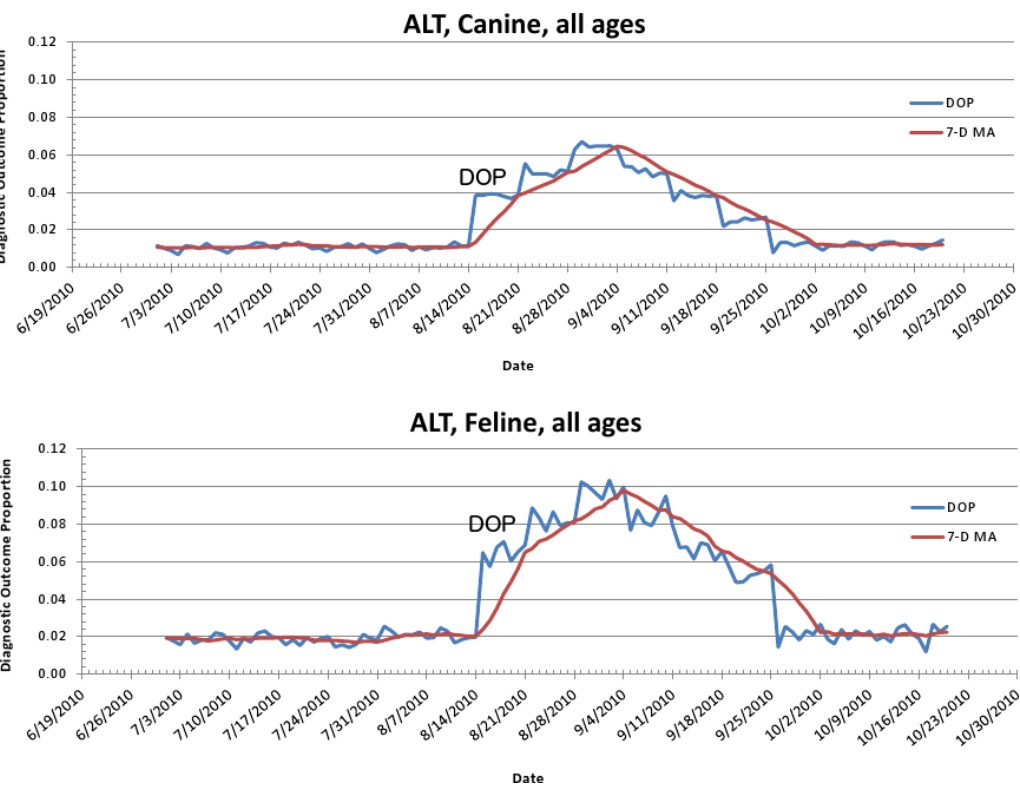

**Figure 10** **Simulated canine aflatoxicosis outbreak using elevated ALT as the syndrome over time.**
Graph shows diagnostic outcome proportions (DOP); red line shows seven day moving average (temporal analysis).

## DISCUSSION

The use of the *PDOR* as the basis for syndromic surveillance in the current study was effective in detecting two distinct and disparate simulated foodborne outbreaks in companion animals seen at a national network of veterinary hospitals. Alerts were generated by the surveillance system for both outbreaks on the actual day they began and, in the case of the aflatoxin outbreak, for the duration of the challenge period; in the infectious agent outbreak, the alerts were generated continuously for 59 of the 64 days of the outbreak, abating five days prior to the end of the challenge period. If these had been real outbreaks, these data would have made it possible to initiate investigations within days of their onset.

The methods developed in this study to detect epidemics differ from the pattern recognition approaches of machine learning and artificial intelligence (although the goals are the same), and are adapted from classical epidemiologic methods for studying patient outcome-related data. The *PDOR* procedure provides a readily interpretable epidemiological measure for quantifying the magnitude of an effect. For example, a *PDOR* of 3 can be interpreted as a three-fold increase in a Diagnostic Outcome Proportion (DOP) at a point in time compared to a baseline time or period. An additional advantage of comparing proportions instead of counts is that a proportion accounts for variation in the

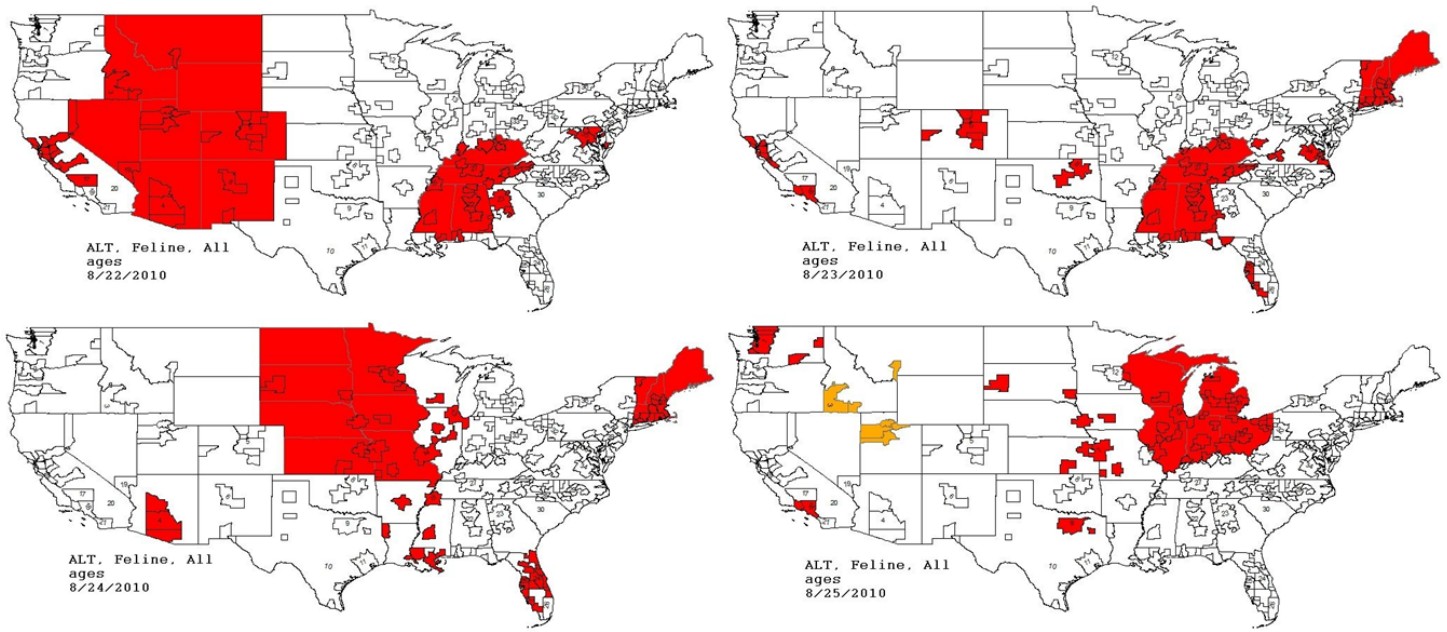

**Figure 11  Spatial analysis of simulated aflatoxicosis outbreak in the US over four successive days.** The most affected US Census Divisions vary by day, indicating the outbreak is occuring on a national level.

number of hospital visits. Measures often used to accompany such statistics, including *p*-values and likelihood ratios, are not generated with this approach. Rather, confidence limits are used to quantify precision, and unlike *p*-values, are interpretable as effect measures.

The five-level alert system used in our approach considers multiple data-generated output measures. The scores are based on the user-defined importance of each of the output measures and can be modified, based on experience, to calibrate the alert system. Users also have the option of customizing how variables used to estimate *PDOR*, such as the lag time and baseline period, are defined. For instance, a user can select different lengths of lag time between the current date and the last day of the baseline time window. We employed a 90-day lag time for foodborne outbreak surveillance because we expect this type of outbreak to be gradual in onset. However, with appropriate adjustments, the *PDOR* procedure is suitable for the surveillance of acute events as well as gradual outbreaks.

The PMR and PRR estimates can serve as the basis for case-control studies that estimate the mortality odds ratio and the reporting odds ratio, respectively (*Miettinen & Wang, 1981*; *Rothman, Lanes & Sacks, 2004*). Likewise, if individual-level food consumption information is available in the medical record, the *PDOR* can be adapted to estimate incidence rate ratios by creating a similarly adaptive study design: a diagnostic outcome case-control study. This bears similarity to a study of proportionate diagnostic outcomes, except that comparison diet and geographic groups must be selected for reasons believed a priori to be unrelated to the diet and geographic region of interest. This allows unbiased estimation of the incidence rate ratio using the diagnostic outcome odds ratio. This study design approach will potentially be effective so long as the source of disease in a particular diet is not present in all diets and in all regions represented in the hospital

**Table 5 Numbers used to simulate the second aberrant event: an outbreak of hepatic disease caused by aflatoxin contamination.** Normal average daily prevalences of anorexia, depression, and ALT elevation are 2.5%, 3.5%, and 2.5%, respectively.

|  | Week | Percent of cats with syndrome caused by aflatoxin | Total percent of cats with syndrome | Total number of cats per week with syndrome at each hospital |
|---|---|---|---|---|
| Anorexia | 1 | 3 | 5 | 2 |
|  | 2 | 5 | 7 | 3 |
|  | 3 | 7 | 9 | 3 |
|  | 4 | 5 | 7 | 2 |
|  | 5 | 3 | 5 | 2 |
|  | 6 | 1 | 3 | 1 |
| Depression | 1 | 5 | 9 | 3 |
|  | 2 | 7 | 11 | 4 |
|  | 3 | 10 | 14 | 5 |
|  | 4 | 12 | 16 | 6 |
|  | 5 | 10 | 14 | 5 |
| ALT elevation | 6 | 7 | 11 | 4 |
|  | 1 | 4 | 6 | 2 |
|  | 2 | 6 | 8 | 3 |
|  | 3 | 8 | 10 | 4 |
|  | 4 | 6 | 8 | 3 |
|  | 5 | 4 | 6 | 2 |
|  | 6 | 2 | 4 | 1 |

network's database (consistent with standard methodologic criteria for control selection in case-control studies).

A limitation to the indiscriminate use of surveillance arises from false positive and false negative errors. Type I (false positive) errors occur when the decision is made to investigate a cluster of aberrant events that are either not attributable to any single set of factors or are attributable to factors that are unmeasurable or beyond intervention. Although all syndromes and diseases have causes, not all causes can be investigated, and not all outbreaks justify investigation, so the costs and benefits must be weighed when deciding whether or not to investigate. Decisions to move from surveillance to investigation must be carefully made by an appropriate team of medical and epidemiological experts. Such investigations can potentially incur considerable expense and investment of personnel (including hospital) and resources. Type I errors therefore lead to unnecessary and unwarranted investigations. A Type II (false negative) error arises when an epidemic occurs, but is either not detected or not investigated. In the face of an actual epidemic, when such signal- to-noise ratios may not be strong, the methods used in the current study can improve surveillance sensitivity by examining strata of important factors, such as geographic location and age. Such efforts to calibrate surveillance instruments require an understanding of veterinary medicine and cannot be relegated to computer algorithms alone.

**Table 6** **Numbers used to simulate the second aberrant event: an outbreak of hepatic disease caused by aflatoxin contamination.** Normal average daily prevalences of anorexia, depression, and ALT elevation are 4.3%, 1.9%, and 2.7%, respectively.

|  | Week | Percent of dogs with syndrome caused by aflatoxin | Total percent of dogs with syndrome | Total number of dogs per week with syndrome at each hospital |
|---|---|---|---|---|
| Anorexia | 1 | 5 | 9 | 13 |
|  | 2 | 7 | 11 | 15 |
|  | 3 | 10 | 14 | 20 |
|  | 4 | 12 | 16 | 22 |
|  | 5 | 10 | 14 | 20 |
|  | 6 | 7 | 11 | 15 |
| Depression | 1 | 3 | 5 | 7 |
|  | 2 | 5 | 7 | 10 |
|  | 3 | 7 | 9 | 12 |
|  | 4 | 9 | 11 | 15 |
|  | 5 | 7 | 9 | 13 |
| ALT elevation | 6 | 5 | 7 | 10 |
|  | 1 | 4 | 7 | 10 |
|  | 2 | 6 | 9 | 12 |
|  | 3 | 8 | 11 | 15 |
|  | 4 | 6 | 9 | 12 |
|  | 5 | 4 | 7 | 10 |
|  | 6 | 2 | 9 | 7 |

In summary, the *PDOR* method provides investigators with a readily interpretable, flexible, and useful tool for detecting disease outbreaks. The ability to customize the various settings and alert levels makes this tool suitable for detection of a multitude of scenarios of disease occurrence. The next logical steps in the application of the *PDOR* methods would be for the detection of actual disease outbreaks using hospital record data, both retrospectively and in real-time, as well as extending the methods to conduct immediate post-detection etiologic investigations utilizing exposure (i.e. dietary) information in the medical record.

**List of abbreviations**

| | |
|---|---|
| **ADORE** | Aberrant Diagnostic Outcome Repository in Epidemiology |
| **BW** | Baseline time window |
| **CL** | Confidence limits |
| **DOP** | Diagnostic outcome proportion |
| **MA** | Exponentially weighted moving average |
| **IRR** | Incidence rate ratio |
| **IT** | Information technology |
| **MSA** | Metropolitan statistical area |
| *PDOR* | Proportionate diagnostic outcome ratio |

| PMR | Proportional mortality ratio |
| PRR | Proportional reporting ratio |
| US | United States |
| USCD | United States census division |
| UC | University of California |

## ACKNOWLEDGEMENTS

The authors would like to thank Drs. Sharon Hopkins, Patrick Sullivan, and Lance Waller for developing the simulated outbreaks used to demonstrate proof-of-concept of the methods in this manuscript, and Dr. Rosalie Trevejo for technical assistance.

### Funding

Funding for this project was provided by Mars Petcare, a Division of Mars Inc., Award # SLO001/UCD Project #200911155. The funders had no role in study design, data collection and analysis, decision to publish, or preparation of the manuscript.

### Grant Disclosures

The following grant information was disclosed by the authors:
Mars Petcare, a Division of Mars Inc.: #SLO001/UCD, #200911155.

### Competing Interests

Philip Kass is an Academic Editor for PeerJ. Amy Hille, Max Sydow, Elizabeth Lund, and Peter Markwell are employees of Mars, Incorporated, the company that provided all funding for this research.

### Author Contributions

- Philip H. Kass, Hsin-Yi Weng and Mark A.L. Gaona conceived and designed the experiments, performed the experiments, analyzed the data, contributed reagents/materials/analysis tools, wrote the paper, prepared figures and/or tables, reviewed drafts of the paper.
- Amy Hille, Max H. Sydow, Elizabeth M. Lund and Peter J. Markwell conceived and designed the experiments, performed the experiments, implemented data transfer, troubleshot errors in the database, reviewed drafts of the paper.

### Animal Ethics

The following information was supplied relating to ethical approvals (i.e., approving body and any reference numbers):

This study did not involve any experimentation, so no institutional review board approval was required. All data was taken from electronic medical records following approval by Mars Petcare, Inc., which owns the data.

## Data Availability

Data was owned/provided by Mars Incorporated. The data is derived in part from a hospital database containing patient information, and the hospital has not given permission to publish this as part of the manuscript.

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
