# Peer review of "Syndromic surveillance in companion animals utilizing electronic medical records data: development and proof of concept"

_PeerJ, doi:10.7717/peerj.1940_

## Round 0.1 · original submission · Major Revisions

Please review all comments raised by reviewers and provide a rebuttal letter that contains a point-by-point response for each reviewer that discusses how and where the issue was addressed in the revised manuscript.

Reviewer 1 ·

Basic reporting

I found the article to be unclear and difficult to read. It was partly due to style. There were some long sentences (e.g. lines 61-68) and too many unnecessary words (e.g. lines 58-61). There were some grammatical errors and the structure of sentences could be improved. For example line 160 ends with 'is'. This is not strictly incorrect but could be improved. Please have this paper reviewed by an experienced editor.
The second issue was that the introduction did not adequately set the stage for the remainder of the paper. The background highlighting examples of companion animal disease surveillance can be much shorter and more time describing the current knowledge on event detection. Remove any information that is not relevant to this study (e.g. lines 79-94). This paper is about using a measure of association rather than case counts for disease detection; this should be the focus of the introduction. Is there any other literature using similar methods?
There were also some incorrect or unjustified statements, some examples:
Lines 53-54 An outbreak investigation does not begin with a case control or cohort study.
Line 120 Unbiasedly - An awkward word and how can you claim there is not bias?
Line 132 - remove the word unbiased
Line 423 - P-values are used in hypothesis testing and dichotomize results as significant or non-significant; a p-value does not quantify effects. Estimation and confidence intervals are more useful by providing a range of values. I do not agree that they can be directly used to measure 'medical impact' - whatever it is meant by that term.
In your formula line 233 please define 'l'.
The descriptions and column labels in Tables 1, 2 and 3 need to be improved.
Figure 2 is not sufficiently described.

Experimental design

Lines 143-146 - Is this an appropriate assumption based on what is known about hospital visits during an outbreak?
Lines 151-155 - Are there any references describing the representativeness of your hospital-based sample?
Lines 156-160 - I don't understand the point being made here.
It was an interesting idea to use a measure of association rather than case counts or proportions of submission data for the statistic in the time series. Control charts including EWMA are based on the Central Limit Theorem. Please comment on the validity of using a measure of association with time as the exposure and a time series control chart. As well using geographic regions as the exposure and then cluster detection using the same geographic regions.
I did not understand your alert levels. Please clarify.

Validity of the findings

I can not comment on the validity of the findings as I need the methods clarified.

Additional comments

You have described a novel approach to disease detection that I feel is worth exploring. However, it was too challenging a read and it needs strong editorial support.

Reviewer 2 ·

Basic reporting

Figures – The figures need a title in the axis.

Experimental design

Major concerns:
- The manuscript lacks references throughout, particularly in the field of syndromic surveillance in veterinary medicine.
Minor points:
Lines to 49-55 – needs references.
Line 70 – needs references.
Lines 108-109 – How representative is this number in the US “hospital” or “animal” population.
Lines 141-143 – The letters in the formula need to be explained.
Lines 151-153 – This needs references and clarification, since it could be argued in the opposite direction.
Lines 165-167 – Define GAP analysis or at least provide a reference. Further explain how the method was implemented.
Lines 167-168 – Clarify “selected for further study, from which the following 10…”. Explain the selection process.
Line 244 – Why the 80% confidence interval?
Lines 289-292 – This is a repetition from what was written in the temporal cluster section, and does not make sense here – time versus space.
Lines 306-309 – Explain how levels were created and indicate the cutoffs.
Line 317 – The scoring system needs to be further explained. It’s not clear how the scores were determined.

Table 1 and 2 – Same comments as above.

Validity of the findings

Major concerns:
- Since both outbreaks were induced, it is not clear how sensitive would the system be to a real outbreak, and how much time would take to respond. Although, this manuscript aims to develop and proof a concept, this should argued in more depth in the discussion.

Additional comments

Strengths of this manuscript:
1. Develops a novel approach to an extremely important problem, that can be applied to other populations in other locations.
2. Presents a method that can be very useful in developing an early warning system and in preventing disease to spread to a broader population.

Reviewer 3 ·

Basic reporting

No Comments

Experimental design

The experimental design is broadly well described but some specific details should be clarified prior to publication:

1. The mechanism of outbreak insertion and nature of inserted data should appear in the methods not the results with more information on how the inserted data was modeled (ie how the number of animals exposed and number affected were arrived at) and also how the inserted data was made consistent with other records received at that time. For instance , how were lethargy and depression encoded in the record and subsequently detected as lethargy.

2. Can you detail the nature of clinical records collected i.e. did you transfer all data captured by the hospital record system or just consultation records or a mixture of records and lab results etc etc - is there a standard format to the records collected at all hospitals. You mention that the hospitals concentrate on preventative medicine does this mean a very high percentage of consults are "well animals" can you give numbers which will help justify assertions regarding constant numbers of unaffected cases etc.

3. It would be worth commenting on the mechanism of owner consent to allow use of their data in this study

Validity of the findings

The data shows a clear detection of the injected outbreak data

Additional comments

The paper is a fascinating report of a new and important initiative in developing syndrome/disease surveillance in the veterinary small animal sector with an initial proposal for how this might be implemented through a two stage alert system allowing identification of outbreaks.

General points

Line 163 - The workshop was convened from experts in various disciplines - was this an internal panel or external panel - whilst not a critical factor, it would illustrate the methods employed to ensure the most appropriate panel.

Line 165 While a set of syndrome for subsequent analysis were identified by this panel, I’m not clear why the subsequent analysis of what was actually pragmatically available to analysis was a “Gap Analysis”. Gap analysis appears to have a more specific definition than simply defining the difference between what we want to analyse and what we can analyse. If this was a true GAP analysis the methodology should be described or referenced.

Line 167 - Would it more appropriate to refer to all the “syndromes” as clinical findings. Syndrome would more often imply a complex of clinical findings that might be associated with disease. Eg. Diarrhoea alone might best be described as a clinical finding where a complex of diarrhoea, raised ALT and anorexia could be described as a syndrome (with a likely association with liver disease) - this is an issue for all publications in this area.

Line 186-188 - This might be best explained with a flow diagram - I assume the data was transferred as files via FTP and then the local scripts imported this into your MySQL database but it is not clear. Was the transferred data in a structured format (XML, JSON, CSV) with some form of schema or coded in a manner that allowed unambiguous transfer of information into you database.

Line 190. What is “loader” ? is it the person uploading the data from the hospital network ?

Table 1: Legend and column headings are not well structured and while some explanation is provided in the text, the table is hard to follow.

Line 149. When you state a syndrome is rare - can you suggest a figure here. i.e. can you stipulate a level of occurrence over which your assumption fails to hold.

156-160 Regarding your lesser assumption (proportion of patients without syndrome appearing constant over time), you assert this appeared to be true - can you validate this. seasonality may well occur both in syndromes (eg diarrhoea) and presentation of animals for other diseases (eg skin disease) and non-sickness related consultations (routine/elective work may be less common around school holidays for instance)

Line 245 Consultation numbers might be expected to be very different throughout the holiday season (perhaps comprising far more sick animals) is it enough simply to exclude Christmas day and simply set back the baseline one day.

Line 248 The abbreviation LCL is not declared in the text elsewhere (?Typo)

Line 299-300 Can you explain why you set these thresholds (analysis of outcomes had different thresholds been chosen)

Line 353 - What response team ?. no methodology for the management of outbreaks is detailed.

Line 380 - Why do graphs only show ALT data especially given that anorexia was a stronger signal?

Figures generally - date axes are messy and hard to read . .

Figure 6 - Salmonellosis not mentioned anywhere else in manuscript

---

## Round 0.2 · Minor Revisions

Although the reviewers were pleased with changes made by the authors, there are still a few issues that need to be resolved. Please provide a rebuttal letter that lists each point raised by reviewers and how and where this was addressed in the revised manuscript.

Reviewer 1 ·

Basic reporting

• This version of the paper was clear and an easier read.
• I still find the detailed description of outbreaks in companion animals in the introduction not relevant to the content of the paper. More background on using proportions from an epidemiological measure of effect compared to using the more usual count or rate data for event detection would be useful and interesting.
• Some minor points:
o Line 211 and Figure 1 - petient? Or patient.
o Figure 5 - PAC = Pacific?
o Figures 6 and 10 - MA or EWMA?

Experimental design

• Research question was well defined and meaningful.
• I was better able to understand the methodological reasoning on this reading.

Validity of the findings

• Data and statistics were sound.
• Conclusion was clear and connected to the research question.
• Can you comment on using time (or place) as your exposure for measures of effect and then plotting this proportion in time (or place). I understand it was a means of comparing a current event to a baseline but it does sound like a circular argument which will affect the validity of your result.

Additional comments

This is an interesting and novel approach to syndromic surveillance and event detection.

Reviewer 3 ·

Basic reporting

No Comments

Experimental design

I guess I am still not happy with the description of how the outbreaks were 'injected' into the data. Your methods section basically has a black box in it, namely the paragraph (219-223) that says experts designed the outbreak. In lines 360-375 and 394-406 you say what the key clinical finding(s) were but I honestly feel that it would assist the narrative of the paper (and the reader in following it) if you said something like:

For the first outbreak the single clinical sign of diarrhoea was added to (?randomly) selected records (or simulated cases with the clinical sign of diarrhoea were added to the data stream) for patients with selected age categories.

For the second outbreaks, clinical records were inserted in to the data stream with key features of hepatic disease (namely raised ALT, anorexia, depression and icterus). inserted cases were not (or were ?) attributed with all clinical signs but frequencies of signs were modelled according to known patterns of disease (reference)

You might include a table detailing the key features of the inserted data

This would help the paper make more sense to me and without it I do not feel the study could be reproduced by another investigator as set out in the journal guidelines

Validity of the findings

No Comments

Additional comments

This is a really exciting study and will set a valuable benchmark for others working in the field but I am keen that methodologies are more clearly laid out.

Bits and Peices:
Line 322 "PETIENTS" - is the actual name of the table (ie a typo by the database designer) or a typo in the text ?

Line 185-186 - Is the author perhaps pursuing personal dogma in asserting the value of open source in this study - does it actually facilitate any of the requirements for system performance you allude to. In the reviewers experience the key means of getting good software is using what the employed developers are most familiar with. It is unlikely that use of proprietary software will in any way constrain sharing of methodologies with any partners in institutions likely to use it. If the author feels this assertion is valid, perhaps it should be referenced.

Line 187-190 - you assert the system can be envisioned as the centre of a constellation of potential IT sources. Is this more speculation than demonstrated methodology (therefore more appropriate for the discussion) given that you only gathered data from one common medical record software in this study ?

Line 204 - I'm still curious what your flat file format was (CSV, TSV or more structure JSON, XML) and whilst not critical to understanding the paper might inform discussions on data standardisation that might facilitate other studies in this arena (surely the aim of publishing).

---

## Round 0.3 · accepted · Accept

Reviewers are generally now satisfied with changes submitted except for the typo of 'petients' which should be 'patients'. Please correct this.

Reviewer 1 ·

Basic reporting

I prefer patients to petients but perhaps is a small thing,

Experimental design

I clearly understand your methods now.

Validity of the findings

No comments